# A Blockchain-Based Solution for Diploma Management in Universities

**Mihai Caramihai** [1,*] **and Irina Severin** [2]

1 Faculty of Computer Science & Automation, University Politehnica Bucharest, 060042 Bucharest, Romania
2 Faculty of Industrial Engineering & Robotics, University Politehnica Bucharest, 060042 Bucharest, Romania; irina.severin@upb.ro
* Correspondence: m.caramihai@yahoo.com

**Abstract:** Diplomas are of particular importance in society, as they serve as official proof of education. Therefore, it is not surprising that counterfeits of such documents have become common. Thus, employers usually have to verify diplomas manually with the issuer. In line with the principles of social responsibility (accountability, transparence, ethical behavior, respect of stakeholders' interest), ensure inclusive and equitable quality education (SDG 4) become a priority for universities. Blockchain technology creates opportunities to surmount these obstacles because it has revolutionized the way people interact with each other. For that purpose, a total of 147 students from a technical university in Romania answered questionnaires to determine how familiar they are with blockchain concepts and what they think about the implementation of this technology. Further, structured interviews were conducted with education and ICT experts to determine the main advantages and disadvantages, applications, and needs for adopting blockchain technology. Therefore, the objective of this paper is to explore the influence of blockchain on education through the assessment of 69 variables. The final part contains the consequences of the analysis's findings and potential future advancements.

**Keywords:** blockchain; higher education; diploma verification; pilot model





## 1. Introduction: Blockchain-Based Educational Certificates, Fake Diploma Problems

Today's generation has the greatest access to education of all time, and so, one of the most important things about education is quality. As a consequence of this great access, new institutions have increased, institutions that issue undergraduate and postgraduate diplomas. Graduates' employment is still affected by the reputation of the institution where they graduated.

However, as the number of institutions that issue a diploma has grown, the possibility of fake diplomas has also increased. For example, in 2018, media [1] brought the public's attention to the case in which two politicians from Spain forged master's degrees. As a consequence, the institution suffered reputation damage, because it seemed that it helped the politicians in this process. Moreover, the problem of fake diplomas can increase the perception of young adults that the road to success does not need to be the one that requires work, because there is always the possibility of shortening the path and decreasing the effort. Hence, as will be seen below, a blockchain technology implementation would prevent the issuing of fake diplomas with a date in the past, i.e., it would be very difficult or nearly impossible to create counterfeit diplomas with backdated dates. Thus, a technical solution based on blockchain would become useful somewhere in the future [2,3] without having an immediate applicability, which can represent a limitation for those who want urgent solutions.

For the purpose of this paper, by "blockchain" we mean an immutable, unforgeable, decentralized public registry of ownership secured through cryptography [4,5]. Although mainly used for digital currencies, it seems that it could significantly impact healthcare, the pharmaceutical industry, and drug supply chain management [6]. Blockchain technology



helps to conduct verification and checks without relying on third parties. Blockchain data, once added, cannot be altered or deleted because cryptography guarantees data integrity and is dependent on the adjacent completed block [7,8]. Most of the data can be added to the block with trust verification and the transactions registered in chronological order are time-stamped. As the data are kept in decentralized ledgers, the danger of losing data is eliminated [9]. The data are transparent and observable to persons with granted access permissions [10]. The risk of duplicities or fraud is avoided due to the agreement protocols [11]. The education framework requires all these features and benefits. An employer needs to verify and view certificates [12]. The data cannot be lost or altered, and confidentiality is maintained; moreover, a third-party intermediary is not required [13,14].

A student-centered approach demands cost reduction, increased efficiency, instant authentication, transparency, safety, and easy retrieval of stored data. These requirements are all inherent features of blockchain technology, which includes a peer-to-peer network of decentralized and distributed ledgers, providing a secure environment. Transactions are time-stamped and stored in ledgers owned by individual parties with consensus, making it nearly impossible to hack or commit fraud. These features are essential for the education domain, as they guarantee secure and time-stamped transactions and information, easily encrypted and stored for future reference. Certificates are issued with immutable proof and are verifiable by administrators and employers with the student's permission, and authentication occurs with a simple click of a button [14]. By adopting blockchain technology, various benefits can be achieved, including an effortless verification mechanism, accuracy of facts, convenience, cost-effectiveness, time-efficiency, privacy and security, secure trades, transparent and immutable data, and availability. The use of a portal by employers or a third party ensures an easy verification process, while the digitization of certificate creation and validation reduces costs and accelerates the process. Additionally, students' achievements are stored securely on a private database and modifications are approved by an agreed mechanism (i.e., a set of rules, procedures, or protocols that have been established and accepted by relevant parties or stakeholders and any alterations are legitimate, authorized, and transparent), ensuring the data are transparent and immutable. Finally, the confirmation of certificates is available at any time and from anywhere, ensuring accessibility [15]. Some of the issues with implementing blockchain technology in higher education include the following [4]:

1.  Cost: implementing blockchain technology can be expensive, and many educational institutions may not have the resources to do so.
2.  Integration with existing systems: integrating blockchain technology with existing educational systems can be challenging, especially if those systems are outdated or not designed to work with blockchain.
3.  Technical expertise: implementing and maintaining blockchain technology requires technical expertise, which may not be readily available within educational institutions.
4.  Legal and regulatory challenges: there may be legal and regulatory challenges related to the use of blockchain technology, such as data privacy regulations (e.g., the General Data Protection Regulation (GDPR, in the EU), the Health Insurance Portability and Accountability Act (HIPAA, in USA), financial industry regulations, intellectual property laws, cross-border data transfer regulations, etc.).
5.  Resistance to change: Resistance to change can be a significant barrier to implementing blockchain technology in higher education, as stakeholders may be hesitant to adopt new technologies and processes. If one has acknowledged how universities' secretaries are functioning, impactful technical and social issues should be overpassed for implementation, together with sound management commitment.
6.  Limited scalability: blockchain technology can be limited in its scalability, which may make it challenging to implement in larger educational systems.

Fake diploma problems. Schools and universities often face the issue of fraudulent and illegal certificates being awarded to students. Blockchain technology addresses this problem by ensuring that every block in the chain is verifiable. Any instance of fraud is

immediately reported to higher authorities, enabling prompt and decisive action to be taken [4]. Verifying credentials through blockchain technology involves creating a tamper-proof, decentralized record of a student's achievements and storing it in a blockchain. The record, known as a "digital credential", can include information such as the student's name, institution, degree or certificate earned, and any relevant academic or professional accomplishments. To verify a credential, a third party such as an employer or educational institution can access the blockchain network and view the student's digital credential. They can then verify the information against the student's official records and confirm its authenticity.

Individuals without academic qualifications who need employment in a short period of time, as well as unemployed graduates and potential college students, are often the source of prevalent academic fraud. They recognize that intellectual rigor and dedicated study are not essential for attaining their desired outcomes, and unfair behavior can lead to obtaining fraudulent credentials, too. Academic fraud encompasses a wide range of questionable activities in the academic field, such as altering the content of certificates issued by legitimate institutions to meet the needs of the holder. Fake degrees can also refer to certificates obtained from fictitious institutions (diploma mills) that claim to be legitimate but have not been accredited by the stated institution. In addition, intellectual fraud can arise from unaccredited degree-granting institutions, degree mills, and corrupt officials at academic institutions [4].

*Objectives of the Study*

The United Nations' SDGs, or Sustainable Development Goals, are a set of 17 inter-connected goals adopted by the United Nations member states in 2015 [5]. They serve as a universal call to action to end poverty, protect the planet, and ensure prosperity for all. The SDGs cover a wide range of areas, including poverty eradication, quality education, gender equality, climate action, sustainable cities, and responsible consumption and production. Each goal has specific targets to be achieved by 2030, providing a framework for countries, organizations, and individuals to work towards a more sustainable and equitable world.

SDG4 refers to Sustainable Development Goal 4 ("Ensure inclusive and equitable quality education and promote lifelong learning opportunities for all"), which focuses on ensuring inclusive and quality education for all and promoting lifelong learning opportunities.

The key targets of SDG4 include the following:

1. Ensuring that all girls and boys have access to free, equitable, and quality primary and secondary education.
2. Ensuring that all individuals acquire the knowledge and skills needed to promote sustainable development.
3. Promoting equal access to affordable and quality technical, vocational, and tertiary education.
4. Increasing the number of qualified teachers and improving their training and support systems.
5. Eliminating gender disparities in education and ensuring equal access for all.
6. Improving literacy and numeracy skills among youth and adults.
7. Promoting education for sustainable development and global citizenship.

Achieving SDG4 is crucial for fostering inclusive societies, reducing poverty, and promoting sustainable development. It recognizes education as a fundamental human right and a key driver for social and economic progress. By ensuring quality education for all, SDG4 aims to empower individuals, promote lifelong learning, and enable them to contribute positively to their communities and societies.

Based on these aspects, the main objective of this paper is to consider the involvement of blockchain technology in the educational sector, by analyzing 69 independent variables regarding general data, known concepts, aspects important to including blockchain in the educational sector, suitable technologies for education, professions that require blockchain knowledge, benefits, and challenges. The analysis is conducted using the statistical software

SPSS 2019. Finally, the study is a descriptive one, and the data analysis is conducted in a comprehensive manner to provide insights into the meaning of the examined variables.

## 2. Materials and Methods

### 2.1. Analysis of Current State of the Art of European Digital Education Recognition Using Blockchain in Comparison to Best Practices in the World

According to [16], certifications in education are evidence of various aspects: achievement of learning outcomes; the competence that a teacher possesses; the learning process undertaken by a learner; an educational organization or course meeting certain quality criteria; and an accreditation body being authorized to issue certification. The creation of new pathways to education is limited by the existing credential system [10] because people without formal education are challenged in finding jobs (they do not have the credentials to affirm their experience and skills).

In other words, certification is the process by which a certificate is issued as a verification of a claim. The processes involved in a certification are the following [16]: issuing—the claim, issuer, evidence, recipient, and signature are recorded by the institution issuing the certificate; verification—a third party verifies the authenticity of the certificate (by using unique features of the certificate, by contacting the original issuer, by comparing with a centralized database); and sharing—the recipient shares the certificate with a third party (by transferring it, by storing it with a custodian, by publishing it).

There is a set of methods that proves the authentication of a certificate [16]:

1. Identifying/verification: this process entails confirming the identities of the issuer and certificate holder, commonly through identity-validating documents, often involving third parties due to potential intricacies.
2. Standardized processes for issuing and certification: these reveal the approach employed by the issuer in granting the certificate, requiring a standardized methodology to ensure that the recipient fulfills the specified criteria.
3. Mechanisms for regulation and assurance: the trust in the system is bigger if there is a mechanism put into practice to verify that the parties apply the standards in line with the requirements.
4. Security features: essential to authenticating the legitimacy of a certificate, these are achievable through either physical antiforgery measures (signatures, watermarks, unique designs embedded within the certificate) or by maintaining a database of issued claims, enabling third-party verification at any given moment.
5. Accessibility: the certificate's claim should be readily accessible, implying that a copy is retained by the recipient, allowing third-party access upon request by the holder, issuer, or registry; instructions for claim verification are available within the certificate, featuring easily understandable and well-defined information.

In the current society, certificates are limited because a universal format has not been suggested yet [16]. To understand these limitations, several cases will be treated below:

1. Paper certificates exhibit drawbacks, such as the necessity for issuers to maintain a registry of issued certificates for verification, the manual verification processes consuming substantial resources due to the registry, the expensive security features, and the inability to revoke a certificate once issued without the recipient surrendering it.
2. Nonblockchain digital certificates come with the following drawbacks: they can be forged without a digital signature; the lack of standardized digital signatures in many countries limits verification to specific software; electronic records are vulnerable to destruction, especially without backups; and the risk of data leaks is substantial.

Blockchain technology can be difficult to comprehend for educators, learners, and other professional parties. Universities need to know how the adoption of blockchain will affect their privacy, database rights, and other confidential information [4]. One particular new domain of blockchain application is the schooling sector. Areas within the educational domain where blockchain technology can be used are granting qualifications,

accreditation and certifying methods, managing student records and grades, and payments and transactions related to educational processes [17].

The education sector is experiencing a transformation in technology where online classrooms replace the usual classrooms. The current education system does not fulfill the aspirations of the learners. The fees demanded by most universities are so high that the students cannot afford them. The skills imparted to the learners are not promising enough for industry, and most companies need to implement training to the recruits. The traditional education model puts pressure on parents, learners, and society due to the unemployment rate growing across the globe. Hence, adapting a student-centered model, which focuses on the "student" and his needs, is necessary [14]. Furthermore, after completing their studies, graduates occasionally have no access to the academic grading system. In that case, if a graduate cannot access his/her academic certificates, he/she needs to visit the institution personally and request a new document, which can be an expensive and time-consuming process [18].

In the education field, there is a rare presence of some blockchain applications, and the majority of them have a pilot character [14]. The usage of blockchain technology in education is still in incipient phases. Nowadays, only a few institutions use blockchain as a secure platform for validating and sharing personal student data and academic certificates.

The majority of blockchain applications are developed with a focus on the management of certificates, involving handling, storing, issuing, and sharing students' certificates [19]. The following 6 types of applications are used in education and based on blockchain. The first application type is focused on certificate management, and it includes handling credentials, transcripts, certificates, and accomplishment records. The second type is concentrated on competencies and learning outcomes management, by measuring the performance based on qualitative and quantitative evaluations. The third category implies securing a collaborative learning environment, focusing on using blockchain to support the learning environment. The fourth category includes fees and credits transfer, for example, the EduCTX system [18], which transfers credits with the help of tokens. The fifth category relates to digital guardianship consent. The last categories are applications that handle competition management and operations, evaluating professional ability, copyrights management, e-learning systems, examination review, and lastly supporting lifelong learning [20].

The benefits of utilizing blockchain for record storage include enhancing traditional certificate issuing methods, automating certificate processes, facilitating credit transfers, and maintaining comprehensive records of students' accomplishments throughout their educational journeys. Another advantage is that it provides the users with the capacity of verifying the record's validity, and the records are publicly unrestricted and verifiable. Additionally, it allows for decreasing liability issues that sometimes arise while manipulating records [9]. The challenges addressed by blockchain are as follows:

- It offers a way for storing all the data needed by a student during their studies.
- It creates opportunities for carrying fewer documents.
- It creates conditions for changing and storing the personal databases of students.

One significant advantage is that it provides each student a unique ID that allows the student to match up with information. Transparency in viewing grades can be a great advantage [17].

Universities in the world. Owing to the advantages, some universities like Stanford University launched a blockchain research headquarters in 2018. The center is focused on solving the different social and legal challenges of the technology and designing an academic curriculum to expand the use of blockchain technology in diverse fields [20].

In addition, Columbia University cooperated with IBM and established the Center for Blockchain and Data Transparency. The center aims to develop new business models and services and increase the application of these advanced technologies. Similarly, New York University implemented a series of courses on blockchain which cover the emergence

of blockchain, how it works, payment systems, regulatory approaches, smart contracts, tokens, and digital assets [20].

Holberton School is the first institute to adopt blockchain technology. Every student acquires a digital certificate that is within a trustworthy and secured environment where 256-bit encryption and two-factor authentication helps the students in maintaining their privacy. After graduation, everyone's degree is automatically generated after the completion of each course in the study plan. In this way, it was found that blockchain technology contributed to a reduction in fraud with university degrees [8]. Now, some countries and universities are trying to adopt blockchain technology for implementing their privacy policies. Estonia has extensively embraced blockchain technology and aims to achieve advanced outcomes [17].

Best practice analysis. One effective approach for promoting best practices in the education sector is through the creation of outsourced centers, such as innovation hubs, that combine education, research, innovation, and knowledge transfer to facilitate collaboration between business and academic institutions. The use of sandboxes (i.e., an isolated testing environment that enables users to run programs or open files without affecting the application, system, or platform on which they run) for testing and trial runs is also important, as is interdisciplinary collaboration between experts from various fields to better understand and develop blockchain technology. Innovation hubs should also maintain close ties with industry partners and startups to encourage a constructive learning approach, in which students take responsibility for their learning path within projects. Successful universities should provide prestructured online learning paths for students. Positively, blockchain-based education providers are expected to be more agile and decentralized, providing benefits for teachers and students alike [8,15].

Finally, the following several gaps regarding the use of blockchain technology in education have been identified:

One gap is the lack of understanding and awareness of blockchain technology among educators, students, and administrators. This can hinder the adoption and implementation of blockchain-based solutions in education.

Another gap is the lack of standardization and interoperability of blockchain systems, which can make it difficult to integrate blockchain solutions across different educational institutions and systems.

There is also a gap in terms of the scalability of blockchain solutions, particularly in terms of accommodating large numbers of users and transactions.

Also, there are concerns around the regulatory and legal framework surrounding the use of blockchain technology in education, particularly around issues of privacy and data protection.

### 2.2. Analysis of Digital Education Recognition in Romania

The 2020 Digital Economy and Society Index (DESI) report by the European Commission shows that Romania ranks 26th out of the 28 member states of the European Union [21]. The report highlights that one-fifth of the population has never used the internet, and only a small fraction has used it to interact with public authorities. Despite this, Romania is among the top five countries in the European Union in terms of technology graduates, but its employment rate in this field is lower compared to other countries.

Furthermore, two renowned institutions known for their research and innovation efforts, the Executive Unit for Funding Higher Education, Research, Development and Innovation (public institution funding research) and the Polytechnic University of Timisoara, launched the "EBSI4RO: Connecting Romania through Blockchain" project on 1 April 2021, in response to the strategic priorities of blockchain technology in the European Union. The project, which is financed by the European Commission, aims to establish a sustainable ecosystem that promotes the adoption and knowledge of blockchain technology and the European Blockchain Services Infrastructure (EBSI) among citizens, institutions, companies, and government. The project has a two-year implementation period [22]. The EBSI is a part

of the European Blockchain Partnership, which was established in 2018 and has 29 member countries, including Romania. The two partnering institutions plan to set up a new EBSI node that will become operational by the end of the year 2023, in addition to the existing 27 EBSI nodes in Europe. Blockchain technology is used to improve higher education (HE), as well as for the development of an educational infrastructure to support learning. Innovative science learning relationships often involve sustained individual inquiry, intense social interaction with interest groups, and expert mentoring relationships [22]. This research investigates blockchain technology, focusing on the influence of motivation on collaborative work, which positively influences learning performance in higher education institutions (HEIs). In addition, blockchain technology is correlated with decentralization, security and integrity, and anonymity and encryption. It can also be perceived as a consensus mechanism implementation [23].

This paper investigates an alternative in which HEIs include a blockchain network to provide an enhanced sustainable education system. Students' responses were analyzed, and some of them had the opinion that blockchain technology had a very positive influence on learning performance [22].

### 2.3. The Theory of Decentralized Trust and Transparency for Diploma Management in Universities

The theory that can sustain the "Blockchain-based Solution for Diploma Management in Universities" is the theory of decentralized trust and transparency. This theory aligns with the fundamental principles and capabilities of blockchain technology, making it a suitable framework for implementing a solution for diploma management in universities [24,25].

Blockchain technology enables the creation of a decentralized and transparent system where trust is established through consensus and cryptographic mechanisms. It eliminates the need for a central authority or intermediary, relying instead on a distributed network of participants to validate and record transactions securely.

The theory of decentralized trust asserts that trust can be established among participants who may not know or trust each other directly, through the use of cryptographic techniques and consensus algorithms. In the context of diploma management, blockchain provides a tamper-proof and immutable record of credentials, ensuring their authenticity and integrity. This eliminates reliance on intermediaries and enhances trust in the verification process for employers, educational institutions, and other stakeholders [26–28].

The theory of transparency aligns with the transparency features of blockchain technology [29]. Blockchain provides a shared and auditable ledger accessible to all authorized participants, allowing for greater transparency and accountability. The transparency of blockchain ensures that information is open for inspection, making it easier to detect and prevent fraudulent activities [30]. Every participant can verify the integrity and validity of the data recorded in the blockchain [31,32], reducing the need for absolute trust in centralized entities [33]. In the context of diploma management, this transparency enables employers and other relevant parties to directly verify the authenticity and validity of diplomas [34,35], reducing the risk of fraud and sometimes the excessive delays caused by waiting times for authorized entities' verifications and responses [36,37].

By applying the theory of decentralized trust and transparency [38], a blockchain-based solution for diploma management in universities can address the challenges of credential fraud, improve the efficiency of verification processes, and enhance trust among stakeholders [39,40].

### 2.4. Formulation of Research Questions and Data Analysis
2.4.1. Questionnaires (for Students): Justification

To gather information about students' perspectives on blockchain technology applicability, a survey was designed. The survey consisted of 7 sections and was aimed at bachelor, master, and doctoral students at the University POLITEHNICA Bucharest, Romania, Faculty of Automatic Control and Computer Science. All participants were selected randomly to answer the survey; the students possessed fundamental familiarity with blockchain

technology. The intention behind the questionnaire was to formulate a theoretical model for implementing this technology within the educational sector, considering the students as professionals to develop and support the implementation.

Regarding the questionnaire: the first section included 4 questions about the institution, the faculty, the study program of the respondents, and gender.

The second section opened with the year in which respondents heard of the concept and the context in which they heard about it. Also, this section included 18 terms, named variables, which illustrated concepts and whether the respondent knew about them before. The concepts are Smart Contracts, Multi-Signatures, Oracles, Decentralized Storage, Private Key, Validation Process, Blockchain Fork, Hashpower, Proof of Work, Proof of Stake, Block Reward, Wallets, Public Address, Transaction Fees, Blockchain Bloat, Mining, Cryptographic hash function, and Hashtable. The first research question was formulated in connection with the findings from the literature review and was expressed as follows: H1. Did the utilization of blockchain technologies in the education sector influence the demand for blockchain knowledge in various domains?

The third section included respondents' opinions about which aspects need to be considered before including blockchain technologies in the educational sector. There were 9 variables and they were measured on a Likert scale of 1–5. The variables measured were the following: V1. Involvement of Government, strict worldwide regulation, V2. Everything has to be set up with open-source technologies, V3. The ability to get a copy of my own data that can be stored on my own node, regardless of which blockchain system was originally used, V4. The ability to operate a full node and store an encrypted copy of the blockchain used to store credentials, V5. Involving corporations in the process of setting up Blockchain technologies in the educational sector, V6. In-depth education about blockchain technologies for IT professionals and administrative officers in the educational sector, V7. The possibility to process information from various blockchain systems, V8. Clear and transparent rules about who is responsible for payment of fees, and V9. Basic information/education about blockchain technologies for all people involved in the educational sector. Based on the literature review, respectively, the studies, which concluded that basic information about blockchain is essential in education, the second research question was developed: H2. Do considerations to be taken into account prior to integrating blockchain technologies into the education sector impact the adoption of blockchain technologies?

The fourth section included 11 variables regarding technologies which are suitable in the educational sector. The variables were measured on a Likert scale (1–5, where 1 is Not suitable and 5 is Highly Suitable). The variables were as follows: certificates management, competencies and learning outcomes management, evaluating students' professional ability, securing a collaborative learning environment, protecting learning objects, fees and credits transfer, obtaining digital guardianship consent, copyrights management, enhancing students' interactions in e-learning, supporting lifelong learning, and allowing employers and other organizations to view student' educational results and other qualifications on a blockchain. The research question developed was as follows: H3. Did familiarity with blockchain technologies across various professions affect the advantages of implementing blockchain technologies?

The fifth segment encompassed an exploration of professions that demand advanced familiarity with blockchain technologies. The values of these variables were collected using a Likert scale. The professions were as follows: Teacher, Administrative IT-Officer, Administrative Non-IT Officer, Headmaster/Rector/Dean, Educational App-Developer, Researcher in the field of education and educational technologies, and Hardware/software Specialist. The following research question was formulated: H4. What are the professions that need more knowledge on blockchain technologies? Are these Administrative IT Officer and Hardware/Software Specialist?

The sixth section included 9 variables, which illustrate the benefits of blockchain technologies in education. The following benefits were measured on a scale of 1–5: Enhancing learners' activity, Supporting learners' career decisions, Improving management

of student's records, Enhancing trust, Identity authentication, Better control of data access, Enhancing students' assessment, Low cost, and High security. The research question was the following: H5. Is high security the most important benefit?

The seventh section included 9 variables measured on a scale of 1–5 and illustrated the challenges of adopting blockchain in education. The challenges were as follows: Weakening traditional school credentials, Trust, Privacy & security, Cost, Immutability, Scalability, Data unavailability, Setting the boundaries, and Immaturity. The research question was as follows: H6. Are all the challenges mentioned important to consider?

### 2.4.2. Interviews (for Experts): Justification

An interview was developed in order to obtain general information about blockchain technology and the uses of it. It contained 14 questions. The first 3 questions referred to the type of the organization, the domain, and the type of position held (executive or manager). The fourth question pertained to the potential utilization of blockchain technology in higher education, while the fifth question inquired about the pertinent data or units of learning (credentials) associated with blockchain. Next, the sixth question was about the quality assurance standards needed to ensure that the data are accurate, verifiable, and meaningful. The seventh question highlighted the compelling reasons for using blockchain in higher education and the eighth one the most significant hurdles that higher education has to overcome before blockchain can be adopted broadly. The ninth one was about the winners and losers, while the tenth one asked about the participants' opinion about the guarantee that blockchain does not marginalize the population. The eleventh question was supposed to show if blockchain in higher education is just hype and the twelfth question asked who should be involved. The last two questions were about the participants' observations about the adoption of blockchain in general and in education in Romania; the last one included the problem about checking relevant skills when recruiting for a dedicated job or project about blockchain application in higher education.

Through the interviews addressed to the experts, a "correction" of the previous vision (based on students' questionnaires) was desired, in the sense of the transfer of an ideal vision to the real area.

The first research question was as follows:

H1. Is the integration of blockchain technologies essential in higher education?

The subsequent research question was outlined as:

H2. Should higher education institutions address significant challenges prior to embracing blockchain technologies?

The third research question stood as follows:

H3. Is collaboration with companies crucial for the successful implementation of blockchain technologies in higher education?

## 3. Results

### *3.1. Centralization of Data*

#### 3.1.1. Questionnaires (for Students)

Data from filled-in questionnaires were collected from several folders into one data file by using a visual basic algorithm. The role of the algorithm was to loop through each of the three main folders and then to loop through each subfolder to extract the data from the Excel files. In the end, all the questions were saved in an .xlsx file, which contained 147 respondents' responses.

#### 3.1.2. Interviews (for Experts)

The interviews were conducted with seven experts from the field of education, and they were either in executive positions or managers. The interviews had eleven questions which were analyzed with the help of Microsoft Excel. The respondents were able to provide complex answers to all the questions. The answers were synthetized and associated, to provide a broad perspective about blockchain technology in the educational field.

### 3.2. Descriptive Statistics and Qualitative Analysis

The study was conducted on 147 participants, from which one was excluded as an outlier. They were students in bachelor, master, or doctoral programs; 73% of which were in bachelor program. Regarding the gender, 62.3% were men, 22.6% were women, and 15.1% did not specify. Most of the participants, 22.6%, had heard of blockchain in the year 2021, and 19.2% heard of it in 2020. Most of the participants had heard of the concept from cryptocurrencies and others from faculty. Some of the other responses were "true games" (i.e., video games or simulations that attempt to faithfully reproduce reality), "workplace", and "internet". A summary regarding the questions and answers is presented in Figure 1.

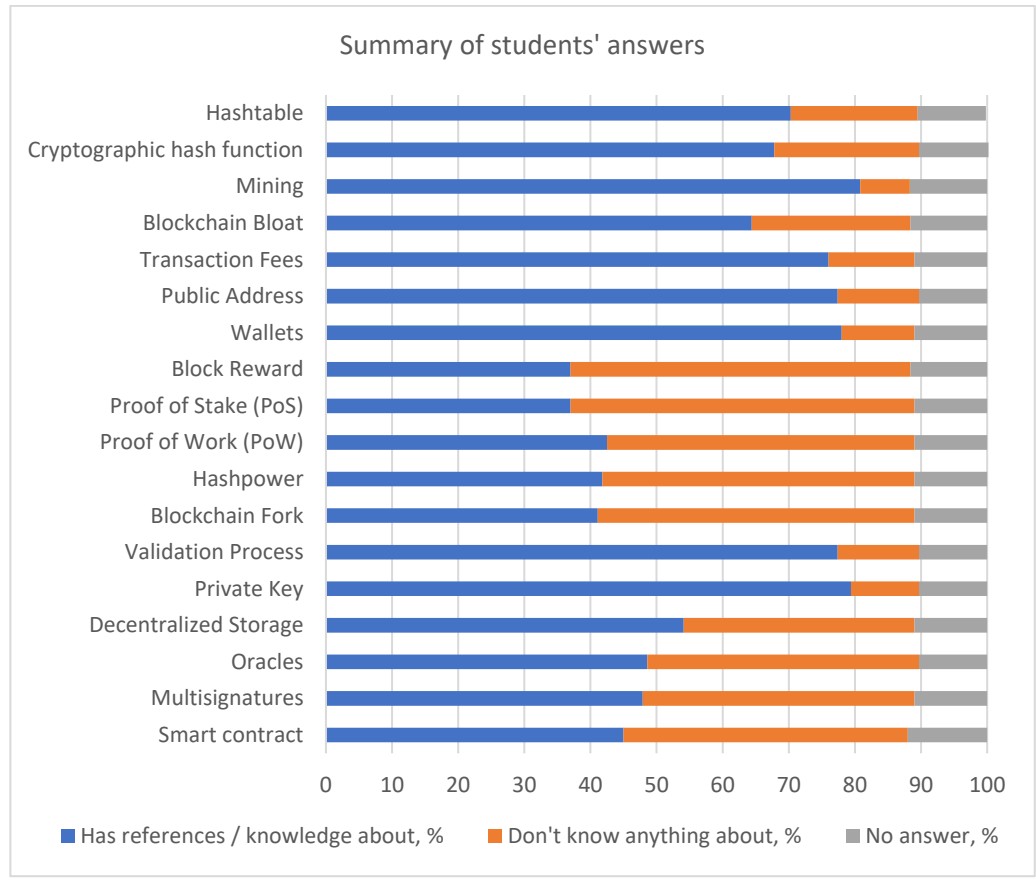

**Figure 1.** Summary of students' answers.

## 4. Discussion

### 4.1. Theoretical Implications

Based on the answers to the questionnaires and in combination with the interviews with experts, the following conceptual/theoretical elements can be identified regarding the relationship between blockchain technology and sustainability in education:

1.  Transparent and immutable record keeping [32,33]: Blockchain enables the creation of a transparent and immutable record keeping system. In the education sector, this can be utilized to securely store and verify academic credentials, certificates, and achievements. By eliminating the reliance on paper-based documentation and central authorities, blockchain technology reduces administrative burdens, minimizes the risk of fraud, and enhances the efficiency of credential verification processes. This streamlined approach promotes sustainability by reducing paper waste and resource consumption.

2.  Decentralized credential verification [34,35]: Blockchain allows for decentralized credential verification, enabling employers, educational institutions, and other stakeholders to directly access and validate educational qualifications without relying on

intermediaries. This increases efficiency and reduces the need for time-consuming and resource-intensive manual verification processes. By improving accuracy and speed of verification, blockchain technology supports sustainability efforts by promoting efficient and reliable hiring processes.

3.  Secure and trusted collaborative platforms [36]: Blockchain-based platforms can facilitate secure and trusted collaborations in the education sector. Through smart contracts, stakeholders can establish transparent agreements and streamline administrative tasks, such as student enrollment, course registration, and certification issuance. By reducing administrative overheads and improving data security, these platforms contribute to sustainability by optimizing resource allocation and reducing the risk of data breaches.

4.  Intellectual property protection [37,38]: Blockchain technology can help protect intellectual property rights in the education sector. Academic works, research findings, and educational content can be securely recorded on the blockchain, establishing proof of ownership and protecting against plagiarism and unauthorized use. By safeguarding intellectual property, blockchain contributes to the sustainability of academic innovation and encourages knowledge sharing while ensuring fair attribution and recognition.

5.  Lifelong learning and skill development [39,40]: Blockchain technology can support lifelong learning initiatives by creating verifiable and portable records of individual skills and competencies. This can empower learners to access new opportunities and career pathways, promoting continuous education and professional development. By enabling individuals to showcase their skills and achievements in a trusted and transparent manner, blockchain technology facilitates sustainable economic growth and adaptability in the rapidly changing job market.

In summary, blockchain technology has the potential to enhance sustainability in the education sector [41] by promoting transparency, efficiency, and security in record keeping, credential verification, collaborative platforms, intellectual property protection, and lifelong learning initiatives. By leveraging the benefits of blockchain, educational institutions can contribute to a more sustainable and resilient education ecosystem.

*4.2. Practical Implications*

In order to have a summative measure regarding a statistical analysis, a variable was created with the scope of knowing the overall percentage of the knowledge of the participants on the topic. The calculated value was 65.03%.

When analyzing the aspects to consider including blockchain technologies in the educational sector, the study used nine variables summarized in the Table 1, considering the scale from 5 (very important) to 1 (not important at all).

Analysis of the variables with SPSS (see Table 2) concluded that the most important aspect is Clear and transparent rules about who is responsible for payment of fees (4.55/5), followed by Basic information/education about blockchain-technologies for all people involved in the educational sector (4.40/5). These answers reflect, on one hand, that students expect transparency in how educational fees are handled; they may be concerned about hidden fees, unclear payment processes, or unexpected costs associated with their education. On the other hand, they reflect that, within the educational sector, an interest in blockchain technology is highlighted: blockchain has the potential to make change in various aspects of education, including credential verification, secure record keeping, and even digital currencies for tuition payments.

**Table 1.** Variable notations.

| Notation | Variable |
|----------|----------|
| V1 | Involvement of Government, strict worldwide regulation |
| V2 | Everything has to be set up with open-source technologies |
| V3 | The ability to get a copy of my own data that can be stored on my own node, regardless of which blockchain system was originally used |
| V4 | The ability to operate a full node and store an encrypted copy of the blockchain used to store credentials |
| V5 | Involving corporations in the process of setting up Blockchain-technologies in the educational sector |
| V6 | In-depth education about blockchain-technologies for IT-professionals and administrative-officers in the educational-sector |
| V7 | The possibility to process information from various blockchain-systems |
| V8 | Clear and transparent rules about who is responsible for payment of fees |
| V9 | Basic information/education about blockchain-technologies for all people involved in the educational sector |

**Table 2.** Mean report.

|        | VI    | V2    | V3    | V4    | V5    | V6    | V7    | V8    | V9    |
|--------|-------|-------|-------|-------|-------|-------|-------|-------|-------|
| Mean   | 3.14  | 3.82  | 4.12  | 3.92  | 3.31  | 4.33  | 3.94  | 4.55  | 4.4   |
| N      | 130   | 132   | 132   | 116   | 130   | 131   | 130   | 129   | 131   |
| StdDev | 1.180 | 1.047 | 0.900 | 1.014 | 1.225 | 0.872 | 0.963 | 0.760 | 0.865 |

The last aspect considered was Involvement of Government, strict worldwide regulation with a mean of 3.14. This means that, in some cases, students may perceive government involvement and strict regulations as potentially leading to bureaucracy, excessive control, or unnecessary restrictions, which could negatively affect the educational system.

The reason why 4.55 is considered more significant than 3.14 is related to the rating scale used in the study. On a typical scale from 1 to 5, a score closer to 5 indicates a higher degree of importance/satisfaction in the participants' responses, while a lower score reflects a lower degree of these aspects.

For most variables, between 129 and 132 of the total 146 responses were included in the analysis, except for V4 where only 116 were included, which was caused by incomplete or missing data from the respondents.

While examining the applications associated with blockchain technologies in the educational sector, the variables were designated as S1 through S11.

The codifications are explained in the following Table 3, with the scale from 5 (very important) to 1 (not important at all).

The most suitable characteristics are Fees and credits transfer (4.02), Copyrights management (3.99), and lastly Evaluating students' professional ability (3.13/5). In our opinion, fees and credits transfer are important for financial planning and academic progress, while copyrights management is vital for academic integrity, career readiness, and navigating the legal and ethical aspects of intellectual property. All of these contribute to a well-rounded education experience for students. Meanwhile, the low score for S3 (Table 4) reflects that students often prioritize immediate concerns, such as completing their education, finding employment, and managing their academic workload. The potential long-term benefits of blockchain verification may not be at the forefront of their minds.

**Table 3.** Variable notations.

| Notation | Variable |
|:---:|:---:|
| S1 | Certificates management |
| S2 | Competencies and learning outcomes management |
| S3 | Evaluating students' professional ability |
| S4 | Securing collaborative learning environment |
| S5 | Protecting learning objects |
| S6 | Fees and credits transfer |
| S7 | Obtaining digital guardianship consent |
| S8 | Copyrights management |
| S9 | Enhancing students' interactions in e-learning |
| S10 | Supporting lifelong learning |
| S11 | Allowing employers and other organizations to view student' educational results and other qualifications on a blockchain |

**Table 4.** Mean report.

| | S1 | S2 | S3 | S4 | S5 | S6 | S7 | S8 | S9 | S10 | S11 |
|:---:|:---:|:---:|:---:|:---:|:---:|:---:|:---:|:---:|:---:|:---:|:---:|
| Mean | 3.83 | 3.43 | 3.13 | 3.8 | 3.8 | 4.02 | 3.7 | 3.99 | 3.65 | 3.57 | 3.36 |
| N | 133 | 131 | 133 | 120 | 133 | 133 | 132 | 132 | 133 | 131 | 131 |
| StdDev | 1.077 | 1.06 | 1.157 | 1.009 | 1.021 | 1.125 | 1.075 | 1.045 | 1.169 | 1.19 | 1.325 |

For most of the variables (S) (Table 4), between 131 and 133 of the total 146 responses were included in the analysis, with the exception of S4 where only 120 were included, which was caused by incomplete or missing data from the respondents.

The research study also analyzed respondents' opinions about which professions require higher knowledge of blockchain technologies, regarding their use in the educational sector. The variables were encoded as follows (Table 5), with the scale from 5 (very important) to 1 (not important at all):

**Table 5.** Variable notations.

| Notation | Variable |
|:---:|:---:|
| Z1 | Teacher |
| Z2 | Administrative IT-Officer |
| Z3 | Administrative Non-IT Officer |
| Z4 | Headmaster/Rector/Dean |
| Z5 | Educational App-Developer |
| Z6 | Researcher in the field of education and educational technologies |
| Z7 | Hardware/software Specialist |

Blockchain-related knowledge is mostly needed for the Hardware/Software Specialist (4.47/5), followed by Administrative IT Officer (4.45/5), and lastly Teacher (3.55/5) and Administrative Non IT- Officer (2.79/5), as resulted using SPSS analysis (Table 6). In our opinion, careers as hardware/software specialists, administrative IT officers, and educational app developers offer students opportunities for job security, innovation, versatility, and the chance to positively influence education and society. These fields are dynamic and aligned with the increasing importance of technology in various aspects of life and learning.

On the other hand, the perceived importance of administrative non-IT officer and teacher roles can vary widely based on individual interests, values, and societal needs. Ultimately, the importance of any career path depends on an individual's passion and skills and the impact they perceive in their chosen field.

**Table 6.** Mean report.

|  | Z1 | Z2 | Z3 | Z4 | Z5 | Z6 | Z7 |
|---|---|---|---|---|---|---|---|
| Mean | 3.55 | 4.45 | 2.79 | 3.55 | 4.34 | 4.35 | 4.47 |
| N | 130 | 132 | 131 | 119 | 131 | 132 | 132 |
| StdDev | 1.295 | 0.794 | 1.093 | 1.133 | 0.771 | 0.838 | 0.842 |

For most of the variables (Z) (Table 6), between 130 and 132 of the total 146 responses were included in the analysis, with the exception of Z4 where only 119 were included, which was caused by incomplete or missing data from the respondents.

Respondents were also asked about the benefits of adopting blockchain technologies in education, by scaling the importance on a scale of 1–5, where 1 is not important to consider as a benefit and 5 is highly important to consider. The variables were encoded as follows (Table 7):

**Table 7.** Variable notations.

| Notation | Variable |
|---|---|
| T1 | Enhancing learners' activity |
| T2 | Supporting learners' career decisions |
| T3 | Improving management of student's records |
| T4 | Enhancing trust |
| T5 | Identity authentication |
| T6 | Better control of data access |
| T7 | Enhancing students' assessment |
| T8 | Low cost |
| T9 | High security |

The most important benefits to consider, as indicated in Table 8, are High Security (4.5/5), Better control of data access (4.25/5), and Enhancing Trust (4.15/5). The last one was Enhancing student's assessment (3.44/5). Hence, students may recognize that data security and privacy are essential for their future academic and professional opportunities. They want to ensure that their educational records are accurate and protected. In contrast, enhancing students' assessment may be perceived as a benefit that, while valuable, may not be of interest for students in the short term. Data security and access control are immediate concerns that affect their current educational experience.

**Table 8.** Mean report.

|  | T1 | T2 | T3 | T4 | T5 | T6 | T7 | T8 | T9 |
|---|---|---|---|---|---|---|---|---|---|
| Mean | 3.65 | 3.65 | 3.85 | 3.89 | 4.15 | 4.25 | 3.44 | 3.49 | 4.5 |
| N | 133 | 133 | 133 | 121 | 129 | 133 | 133 | 131 | 132 |
| StdDev | 1.066 | 1.045 | 1.091 | 1.086 | 0.953 | 0.900 | 1.069 | 1.119 | 0.878 |

The challenges of adopting blockchain technologies were measured on a scale of 1–5, where 1 is not important to consider and 5 is highly important to consider. The variables were encoded as follows in Table 9:

**Table 9.** Variable notations.

| Notation | Variable |
|---|---|
| U1 | Weakening traditional school credentials |
| U2 | Trust |
| U3 | Privacy & security |
| U4 | Cost |
| U5 | Immutability |
| U6 | Scalability |
| U7 | Data unavailability |
| U8 | Setting the boundaries |
| U9 | Immaturity |

The most important challenges to consider, based on variables analysis (Table 10), are Privacy & Security (3.96/5), followed by Cost and Setting the boundaries (3.71/5). The last ones considered were Weakening traditional school credentials (2.84), and Immaturity (3.10/5).

**Table 10.** Mean report.

|  | U1 | U2 | U3 | U4 | U5 | U6 | U7 | U8 | U9 |
|---|---|---|---|---|---|---|---|---|---|
| Mean | 2.84 | 3.76 | 3.96 | 3.71 | 3.35 | 3.58 | 3.47 | 3.71 | 3.1 |
| N | 143 | 144 | 143 | 129 | 140 | 141 | 144 | 143 | 144 |
| StdDev | 1.287 | 1.229 | 1.227 | 1.064 | 1.017 | 1.122 | 1.182 | 1.131 | 1.250 |

Our opinion regarding these answers may be presented as follows:

1.  Privacy and security:

    Increasing concerns: Students are becoming increasingly aware of privacy and security concerns, especially with the growing use of technology in education. They prioritize the protection of their personal information and academic data from potential breaches and misuse.

    Data sensitivity: Educational institutions often handle sensitive data, including student records, grades, and personal information. Any breach or mishandling of this data can have serious consequences.

2.  Cost:

    Financial burden: Cost is a significant concern for many students. Higher education expenses, including tuition, fees, and textbooks, can be a financial burden. Students prioritize finding cost-effective solutions that can make education more accessible and affordable.

    Student debt: The rising cost of education has led to concerns about student debt, which can have long-term financial implications.

3.  Setting boundaries:

    Balancing screen time: With the increasing use of technology in education, students may struggle with balancing screen time. They value the ability to set boundaries between academic work and their personal life to prevent burnout and maintain a healthy work–life balance.

Overwhelm: Students may experience information overload and stress due to the demands of online learning. Setting boundaries helps manage these challenges.

4. Immaturity:

Technology adoption: Immaturity in this context likely refers to the immaturity or lack of readiness of educational technology solutions. Students may face challenges when using technology that is not fully developed or tested, leading to frustration and inefficiency.

Adaptation challenges: Students may find it challenging to adapt to rapidly evolving technology platforms or tools.

5. Weakening traditional school credentials:

Concerns about credential value: While this may be considered a challenge, some students may have more confidence in the enduring value of traditional school credentials. They may prioritize addressing other immediate concerns, such as privacy and cost, over concerns about the long-term value of their credentials.

Hence, the answers reflect the complex landscape of challenges faced by students in the digital age of education. Privacy, cost, technology use, and credential value are all important considerations (Table 10), and students' perceptions of their relative importance may evolve over time.

After analyzing seven interviews with people involved in education, either in executive positions or managers, the results supported the answers to the presented research questions. The respondents highlighted some of the potential applications of blockchain technology in higher education, through certificates, file storage (courses), research preparation, digital identity, currency, and the growth of MOOC. Some of the most important applications mentioned are related to research, as blockchain represents a tool for data sharing and knowledge. The respondents suggested that blockchain can register professional information such as volunteer activities, prizes, and scholarship and can generate them automatically from the block. On the financial level, they said that it can bring funds trackers, payments, grants management, services, credits transfer, or even the tokenization of learning. Blockchain can also help intellectual property and university accreditation and be a big help in document storage.

Respondents said that the relevant data that should be on the blockchain consists of smart contracts, information that is not sensitive, research data, grades, volunteer activities, awards, distinctions, and diplomas. These can benefit from blockchain advantages like transparency, security, and traceability.

Regarding the quality assurance standards that guarantee that the data are accurate, verifiable, and meaningful, respondents said that these qualities are provided by the decentralized database. These are assured by synchronizing across multiple nodes, each having an identical copy. Quality is guaranteed also by the signature of the collector, description of data, and sources, thus making data meaningful and traceable.

Some of the reasons for using blockchain in education are security, functionality, novelty, and transparency. Other reasons provided by the respondents were decentralization, immutability of data, and integrity. Respondents said that by using the latest trends in technology in higher education, blockchain can also inspire new initiatives in research. The most compelling reasons come from the advantages of blockchain technology with multiple applications presented under the first question: self-sovereignty, trust, transparency and provenance, immutability, disintermediation, and collaboration.

Regarding equity, access, and accessibility, governments should have clear politics, the knowledge should be transferred, and risks should be mitigated. When asked about blockchain being just buildup/hype, all the respondents considered that it is not the case and the technology is really improving copyright protection, traceability issues, certificates, and credentials. Looking into the future of blockchain, as time will pass it will improve and awareness of the benefits offered will increase. Respondents think that "in order to go forward and take use of this technology, organizations need to start and implement multiple applications for blockchain, even test them as prototypes to get the first conclusions and

get peer reviewed". Another respondent showed two benefits of using blockchain in the future, manifested through improved efficiency in education with lower costs and better experience and acting in collaborative activities in universities. The stakeholders in this technology in the future will be the government, authorities, academics, companies, and blockchain communities.

Interviewed about blockchain in Romania, respondents said that the country needs more time to understand benefits, more training, and willingness to learn and to know. Although there is no specific national legislation, support, or strategy for blockchain, Romania has a growing number of experienced companies (e.g., Modex, Elrond), innovative startups and projects, and active accelerators—encouraging the adoption of blockchain. Moreover, there are also numerous educational programs including blockchain.

There are a few blockchain pilot projects running in Romania, among them also issuing university diplomas and microcredentials on the European Blockchain Services Infrastructure (EBSI), as one expert underlined. Other respondents showed that Romania evolved in the blockchain landscape, but just in the private sector, and a lot of successful companies in Romania created value in that direction.

The last question involved checking relevant skills when recruiting for a dedicated project about blockchain, and respondents mentioned things like knowledge in the field, case studies, and the use of the systems. Education is also important, along with previous experience, but since blockchain is specific, some skills needed can be cloud computing, web development, formal and informal studies, participating in projects, and cryptography.

*4.3. Synthesis of Results in Comparison to Research Questions*

Based on the results presented in Section 4.2, the following research questions were summarized.

Thus, according to the results summarized in Table 11, the research questions H1 through H5 were supported by results, while research question H6 was not. Nonvalidity of this research question (H6) demonstrates the fact that the analysis of the specialized literature led to the conclusion that certain concerns mentioned are not currently applicable in the educational sector. Concerning research questions H1 through H5, the following can be outlined:

The first research question (H1) formed a connection between the utilization of blockchain technologies in the educational sector and the requirement for specialized knowledge in various professions.

The second research question (H2) verified a relationship between the necessary preconditions for integrating blockchain into educational settings and their subsequent application.

The third research question (H3) symbolized the link between the understanding of blockchain across different professions and the advantages of its implementation.

The fourth research question (H4) made a connection between the prerequisites for introducing blockchain into education and the benefits of such integration, while the fifth research question (H5) established a link between these factors and the challenges faced in implementing this technology.

Consequently, the various components of research questions H1 through H5 are shaped by the presence of several distinct and logical connections, as well as some that are less explicit.

**Table 11.** Summary of answers to the research questions supported by the data.

| Research Question | Conclusion |
|---|---|
| H1. Did the utilization of blockchain technologies in the education sector influence the demand for blockchain knowledge in various domains? | The overall knowledge was calculated with an average of all the components. The knowledge factor was above 50%, actually 65.03%. This means that more than half of the concepts were known by all of the respondents.<br>The research question H1 is supported by the results. |
| H2. Do considerations to be taken into account prior to integrating blockchain technologies into the education sector impact the adoption of blockchain technologies? | The collected values had a mean of 4.40/5, which is more than the validator considered. Basic information about blockchain was extracted from the literature review, and it was a highly mentioned concept in other studies. Also, another relevant concept was The ability to get a copy of my own data that can be stored on my own node, regardless of which blockchain system was originally used, In-depth education about blockchain-technologies for IT-professionals and administrative-officers in the educational-sector, Clear and transparent rules about who is responsible for payment of fees, all with a mean higher than 4/5.<br>The research question H2 is supported by the results. |
| H3. Does familiarity with blockchain technologies across various professions affect the advantages of implementing blockchain technologies? | The technologies that are suitable (over 3.5/5) are Certificates Management, securing collaborative learning environment, protecting learning objects, Fees and credits transfer, obtaining digital guardianship consent, Copyrights management, enhancing students' interactions in e-learning, and supporting lifelong learning. These are 7 technologies out of the 11, which is more than half.<br>The research question H3 is supported by the results. |
| H4. What is the profession that needs more knowledge on blockchain technologies? | The research question H4 leads to the conclusion that the Hardware/Software Specialists (with a mean of 4.47/5) and Administrative IT Officer (4.45/5) are the professions that need more knowledge on blockchain technologies.<br>The research question H4 is supported by the results. |
| H5. Is high security the most important benefit? | High security was a concept mentioned often in the review of the literature. The studies stated that security is the way to look for when adopting blockchain technology. As it was supposed, the results suggested that High security was the most important concept (4.5/5).<br>The research question H5 is supported by the results. |
| H6. Are all the challenges mentioned important to consider? | The challenges are classified as important to consider if their mean is over 3.5/5. This was not the case for Weakening traditional school credentials (2.84/5), Immutability (3.35/5), Data unavailability (3.47/5), and Immaturity (3.10/5). The research question is not supported by results, which shows that some concerns in the literature review were found which were not applicable at the moment, either because those aspects have changed or because they are not relevant in the educational sector.<br>The research question H6 is not supported. |

## 5. Development of a Pilot Model Using Blockchain Concept for "Record Keeping" of Students' Degrees, Certificates and Diplomas Based on the Analysis of the Collected Data: Simulation of a Case Study

Concept and design. The functional requirements include a user management module, students' degrees, certificates and diplomas record keeping, data retrieval, and data validation [42]. The nonfunctional requirements include the efficiency and high availability of record keeping in the blockchain. The model will include the following:

- An application for administrators to login for uploading information into the database. The administrators are the providers of the certificates (universities, accredited centers, or companies that offer a certificate after completing a course) who will be the administrators of the data.
- An application for users (the users will be the students or any third party that wants to verify the authentication of a certificate) will be used, like MetaTask. This is a browser

extension that lets people join their cryptocurrency wallets to websites that require the right of entry to the Ethereum (a cryptographically secure transaction singleton machine with a share-state [42–44]) network. MetaTask has also a mobile application and the users will be able to stay logged in as much as they need in order to find the required information.

- The database will consist of two parts: one relational database and one implemented using blockchain technology. The users and administrators will be able to search in the relational database based on a unique identifier of the certificate. The relational database is necessary [42] because there will be delays in entering data and the searching process if the blockchain database is used.

Development. Firstly, the interface of the application for administrators will be implemented. The languages used are the Bootstrap 4.0 framework and PHP programming language [42,45,46]. The available modules will be add new, update record, save data to blockchain, and verify data from blockchain. The relational database will be implemented using MySQL. The model will operate as follows: An administrator is going to log in the application and, using the add new button, will upload a new certificate in the relational database. Then, he will press the save data to blockchain button in order to store the certificate in the blockchain. If the information is not complete, he can add the incomplete information using add new and then come back and update it using the update record button. To verify a certificate from blockchain, the administrator will need to enter the unique identifier and press the button verify data from blockchain. From the users' side, if someone wants to see a certificate, he will log in into the MetaTask application and enter the unique identifier. This unique identifier will then be used to search in the relational database to check for the correctness of the identifier and, if it exists, it will perform a search in the blockchain (a search that can last a longer time). A proposed structure (with a functionality focused on student demand) is presented in Figure 2:

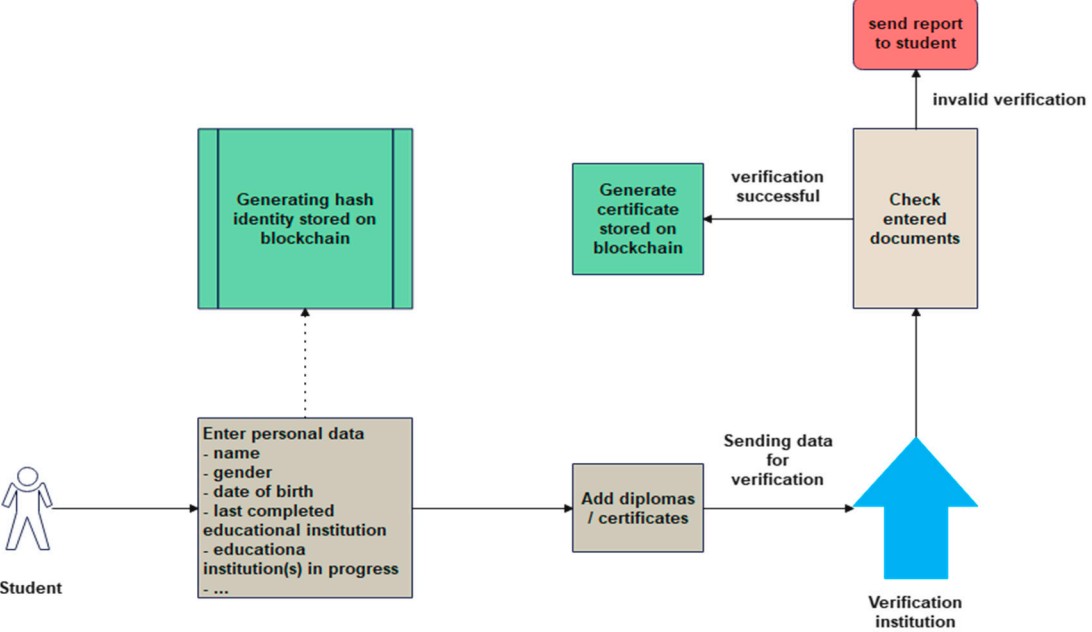

**Figure 2.** Pilot model/student demand.

In the proposed model, the learners are given the responsibility to input all information regarding their education into the system, shifting the task from administrative staff to the students themselves. Consequently, the role of institutions, such as universities, changes from responsibility for data entry to verification. This change in procedure could be quite intriguing for university management, as it represents a significant shift in the allocation of resources and responsibilities. Under the current system, all details about study records are

painstakingly entered by employees, a process that not only consumes valuable time but also incurs substantial financial costs. In contrast, the new approach, where students are responsible for the initial data input and institutions focus on validation, promises to be not only cost-effective but potentially more engaging for students. It might foster a sense of ownership and involvement in their educational journey, enhancing their connection with the educational process. Additionally, it could release administrative staff to concentrate on more complex and value-added tasks, thereby optimizing institutional efficiency and effectiveness.

## 6. Conclusions

Educational institutions and universities that have embraced blockchain mainly use its functionalities to keep and transfer academic documents and certificates. Although the opportunities are promising, issues such as data protection, scalability, and cost provide challenges to the overall adoption of blockchain in the academic sector. Technology is developing continuously, so blockchain might play a bigger role in education. Blockchain has changed the economic drive via cryptocurrencies and coins. This technology would have a changing effect on schooling, simplifying keeping records and sharing them, improving safety and enhancing trust, facilitating the hiring procedure, and providing learners rights to their academic records anytime and anywhere.

Individuals who demonstrate an interest in exploring the future role of blockchain in education have the opportunity to pursue a specialized degree. Delving deeper into the subject of blockchain, through reputable courses focused on the practical applications of this technology, cryptocurrencies, and the ethical and legal considerations associated with its usage, can greatly enhance their educational experience. Such a curriculum is designed to provide researchers and students with the foundational knowledge they need to progress in this rapidly evolving field. By engaging with carefully structured learning modules and interacting with experts, participants can acquire a nuanced understanding of blockchain's potential impact on education and the broader societal context.

Regarding the proposal presented, despite the fact that only five out of six research questions were supported by data, the final results indicate that there are several connections between the use of blockchain technologies in education, the importance of blockchain knowledge in various professions, the considerations that must be taken into account before implementing blockchain in education, the benefits of adopting blockchain technologies, and the challenges that arise when doing so. These relationships exhibit varying levels of intensity, ranging from strong to weak, and influence with different degrees the individual components that make up each research question. This variability underscores the complexity of the connections and highlights the need to consider each one within its specific context, as the strength or weakness of a relationship may provide a unique insight into the underlying dynamics that shape the research questions.

**Author Contributions:** Conceptualization, M.C. and I.S.; methodology, M.C and I.S.; software, M.C.; validation, I.S.; formal analysis, I.S.; investigation, M.C.; resources, M.C. and I.S.; data curation, M.C.; writing—original draft preparation, M.C.; writing—review and editing, I.S.; visualization, M.C. and I.S.; supervision, M.C.; project administration, M.C.; funding acquisition, M.C. All authors have read and agreed to the published version of the manuscript.

**Funding:** This research received no external funding.

**Institutional Review Board Statement:** Not applicable.

**Informed Consent Statement:** Not applicable (anonymous questionnaires have been collected, all students agreed to participate voluntary to fill-in the questionnaires).

**Data Availability Statement:** All data are summarized in tables given in the article.

**Acknowledgments:** This paper was developed in the framework of the Erasmus+ Project 2020-1-CZ01-KA226-HE-094408 entitled "Boosting Sustainable Digital Education for European Universities", implementation period: 2021–2023.

**Conflicts of Interest:** The authors declare no conflict of interest.

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
