# Peer review of "A Blockchain-Based Solution for Diploma Management in Universities"

_sustainability, doi:10.3390/su152015169_

Round 1
Reviewer 1 Report (Previous Reviewer 1)
A Proposal Regarding a Blockchain-base Solution of Diploma Management in Universities is an interesting article that addresses an important issue regarding the use of blockchain in storing and verifying education credentials. The authors comprehensively address the space, including factors such as cost and privacy concerns.
One issue, which is easy to fix, is that the study focuses on Romania but this is not clear at the beginning of the paper. The findings are applicable anywhere in the world, but it is good to inform the reader about the scope early. As such, I recommend adding this detail to the title, abstract or early in the introduction. It is important to be up front with readers regarding the scope of the article.
Finally, on line 36 it states that the implementation of blockchain in issuing and verifying credentials would “prevent the issue of fake diplomas with a date from the past.” Unless retroactively issuing credentials on the blockchain, it seems the system would only work on credentials moving forward, not past credentials. As such, fake historical credentials could be issued off the blockchain, but this would become less of an issue as the population ages.
While the paper can be easily understood, it requires a substantial edit. I recommend the authors either find a native English speaker to carefully edit the paper or use an editing service.
Author Response
The authors appreciate and thank for your feedback and hope that the article’s adjustments and clarifications allow its’ better readability. Thank you for your cooperation !

Reviewer 2 Report (New Reviewer)
Dear authors,
many thanks for the opportunity you gave me to read your manuscript ID sustainability-2551771, titled “A Proposal regarding a Blockchain-based Solution for Diploma Management in Universities". This study investigates the involvement of blockchain technology in the educational sector. The research topic is interesting, but the current version of the manuscript presents several weaknesses to be solved and in many sections the writing is poor, making some sentences difficult to understand. My biggest concern is that the current version of the manuscript does not include a clear research design and methodology is
not rigorous. Besides, the manuscript is not well structured and developed. I suggest to rewrite the manuscript starting from the Research Questions. I suggest to rename headings that contribute to confuse the readers (see Materials and Methods for instance that contains several sub-headings:
2.1. Analysis of current state-of-the-art of European Digital Education Recognition using blockchain in comparison to best practices in the world.
2.2. Analysis of Digital Education Recognition in Romania
2.3. The theory that can sustain the Blockchain-based Solution for Diploma Management in 287 Universities
2.4. Objectives of the study
2.4. (again) Development of hypothesis vs questionnaires / interviews
2.4.1. Questionnaires (for students). Justification
2.4.2. Interviews (for experts). Justification
I suggest you to avoid to dedicate a specific sub heading to the objectives of the study that should be included in introduction sections.
Methodology is unclear. How respondents have been selected? What sampling procedures has been followed?
The hypotheses are not supp
The aim of this study has not been clearly defined and exposed, leaving the reader with many doubts until the end of the paper. Furthermore, the originality of the study, as well as its contribution in advancing knowledge from previous studies regarding the issues introduced,is not clear.
It’s not clear how the hypotheses formulated have been developed and sustained by theoretical background
Results remains unclear.
Discussion should be completely revised. In this section You should critically discuss your findings comparing them with previous studies, while this section include theoretical implications regarding the relationship between blockchain technology and sustainability in education, that is not the main focus of the study and practical implications, that mostly include study’s results. Finally a pilot model for “record keeping” of students’ degrees, certificates and diplomas based on the previous analysis, is proposed, but it is not clear how its has been built.
In conclusions, the manuscript contains several good ideas but in my view it should be completely rewritten and revised before resubmission.
Good luck.
It requires a professional proofreading
Author Response
The authors appreciate and thank for your feedback and hope that the article’s adjustments and clarifications allow its’ better readability. Thank you for your cooperation !

Reviewer 3 Report (New Reviewer)
The paper "A Proposal regarding a Blockchain-based Solution for Diploma Management in Universities" brings a contribution to the sustainability of higher education and its outcomes. The information presented is interesting, the methodology seems to be relevant as well, but more clarification is needed in some occasions. There are a few places where the paper would benefit from modifying the structure, and/or adding further details. One thing which is essential for the whole paper needs to be improved, namely the "hypotheses" design. It should be turned into a statistical "testing hypotheses", or reformulated. Revisions of the language and notation are needed. The notation and terminology should be consistent throughout the paper. The Conclusion should contain more information about the results of the analysis.
Attached to the report you may find the manuscript with highlighted text and reviewer's comments and suggestions. Please pay attention to them and address them properly one by one. It was a hard work to go through the paper and guess what some formulations mean. However, I believe it was worth the time and the paper will make a good contribution to the subject, provided the necessary modifications will be done. Continue doing good job and take the time to address the comments properly.

The quality of English should be improved. Revisions of the language and notation are needed. The text can be read and understood, but some words should preferably be replaced by more suitable ones. Some terms and formulations seem to be a direct translation from author's mother tongue into English. Those should be reformulated in true English. Some suggestions are included in the commented manuscript attached to this report.
Author Response
The authors extend their profound gratitude for your invaluable feedback, recognizing the instrumental role it has played in refining the manuscript. Your insights and cooperative spirit have significantly contributed to enhancing the readability and overall quality of the article. It is our earnest hope that the revisions and elucidations made will resonate with the readers and fulfill the intended purpose. Once again, thank you for your collaboration and thoughtful contribution, which are held in the highest regard.

Round 2
Reviewer 2 Report (New Reviewer)
Dear authors,
thank you for giving the opportunity to read your manuscript again. I am glad that all the comments and the suggestions provided in the previous round have been successfully addressed and the paper is significantly improved. Good luck!
Author Response
Thank you for your appreciations!
Reviewer 3 Report (New Reviewer)
The paper "A Proposal regarding a Blockchain-based Solution for Diploma Management in Universities" brings a nice contribution to the sustainability of higher education and its outcomes. The information presented is interesting, the methodology seems to be relevant as well, but more clarification is needed in some occasions. The paper is well organized and structured. One thing which is essential for the whole paper needs to be improved, namely the "hypotheses" design. It should be reformulated as "research questions" and the conclusions should be modified accordingly. Some revisions of the language and notation are still needed. The notation and terminology should be consistent throughout the paper. The Conclusion might contain more concrete information about the results of the analysis.
Attached to the report you may find the manuscript with highlighted text and reviewer's comments and suggestions. Please pay attention to them and address them properly one by one. It was a hard work to go through the paper and reflect about improvements. However, I believe it was worth the time and the paper will make a good contribution to the subject, provided the necessary modifications will be done. Continue doing good job and take the time to address the comments properly.

The quality of English should still be improved. Revisions of the language and notation are still needed. The text can be read and understood better, but some words should preferably be replaced by more suitable ones. Some suggestions are included in the commented manuscript attached to this report.
Author Response
The answers are attached

Round 3
Reviewer 3 Report (New Reviewer)
The paper "A Proposal regarding a Blockchain-based Solution for Diploma Management in Universities" brings a nice contribution to the sustainability of higher education and its outcomes. The information presented is interesting, the methodology is relevant and well described. The paper is well organized and structured. Some minor revisions of the language are needed.
Attached to the report you may find the manuscript with highlighted text and reviewer's comments and suggestions. Please pay attention to them and address them properly. Thank you for collaboration, you did a great job.

The quality of English is good, only minor revisions of the language an punctuation are needed. The text can be read and understood well, but some words should preferably be replaced by more suitable ones. Some suggestions are included in the commented manuscript attached to this report.
Author Response
The reply is attached
This manuscript is a resubmission of an earlier submission. The following is a list of the peer review reports and author responses from that submission.
Round 1
Reviewer 1 Report
This paper presents interesting data on using blockchain to issue diplomas. The paper, however, drifts between blockchain related topics and requires a substantial revision to bring coherence to the argument. The core contribution of the article is the survey and expert interviews. These illustrate the level of knowledge students at one university have with respect to blockchain and how this would impact the adoption of blockchain to issue diplomas. However, the title of the paper suggests a “proposal” and the abstract a “case study”.
After this, the paper provides a great deal of information that is not directly related to a proposal for blockchain diplomas. For example, from line 185 the authors talk about 12 types of applications where blockchain is used in education. Only one of the 12 was related to credentials. Further, the universities around the world section (line 209) discusses research centres at several universities where credentials are not being issued. However, Hoberton (line 229) appears to be directly on point but only receives 7 lines. If making a proposal, it would be important to know exactly how Hoberton issued the diplomas and the overall impact. Did it work well? Any unforeseen consequences?
The beginning section defining the problem includes examples such a Spanish politician being assisted by a university of obtain fake credentials (line 29) and problems with corrupt university employees. While it is true that the blockchain cannot be altered, use of the blockchain would not necessarily prevent corrupt officials from issuing fake diplomas. It would simply prevent them from issuing fake diplomas with a past date. It would also take a generation to realise this benefit as a market for fake diplomas will continue to exist with diplomas dated prior to the adoption of blockchain issued diplomas. This does not negate the argument of the paper, but is a limitation that should be mentioned.
Privacy is a significant concern of blockchain based credentials that needs to be better addressed if the authors decide to continue with a proposal. Hacking is also a significant concern depending on how the credentials are issued. A secure blockchain cannot be hacked. However, the issuer of the credentials can be hacked.
These are different way the authors could make the paper more coherent. My suggestion is to change the goal from a proposal to identifying the knowledge gaps that need to be addressed for students and experts to be comfortable with blockchain issued credentials. The authors have already identified these through their research and this approach does not require defending every aspect of a proposal. I also suggest the authors make it clear in the title that this paper is focused on Romania and provide details for how diploma verification/transcript verification works in Romania for readers in other countries. The authors touch on this in beginning of the paper but don’t provide details for Romania.
Author Response
The authors appreciate and thank for your feedback and hope that the article’s adjustments and clarifications allow its’ better readability. Thank you for your cooperation !

Reviewer 2 Report
The have gone through the article, and it is interesting. But I found a few minors suggestion/corrections. please note it
1. the introduction part should be more readable.
2. The literature review gaps should be more clear
3. The validation part is missing.
4. Language is more concern
5. references should be aligned with the text and references list.
Author Response

(The authors gave the same response as above.)

Reviewer 3 Report
In the paper, the authors presented the results of the questionnaires among the student about the basic terms used in blockchain technology and interviews with SEVEN experts from SEVEN different roles (see Table 4.7). It should be mentioned that, as could be learned from the abstract (see lines 11-2), two contributions from this article are expected:
- 1) application of blockchain technology in the field of higher education (also the title of the paper refers to this)
- 2) find out to what extent higher education is ready for BC adoption
While authors claimed that the study's objective (lines 308-316) is to analyze the involvement of blockchain technology in the educational sector. However, how the beforementioned outlined contributions and defined study's objective are co-related authors is not described. Moreover, the authors didn't propose applying blockchain technology in higher education, as stated in the title and abstract of the paper. Conditionally interviews could be seen as the only method to achieve the objective - however, the population (i.e., only seven different experts from different roles in HEI) requires to be wider for any conclusions to be made. Another major problem of the paper is its organization and presentation itself.
- The introduction is hard to read; different terms and concepts are thrown into, without any meaningful correlations/connections between paragraphs. For example, lines 37-42 and 49-60 discuss the same topic; the presentation is unstructured and unclear; references used within the text are not recent; a lot of concepts (lines 64-76) are again just thrown into, without any proper description to be set in the context of the paper.
- Backgrounds including clear process description on how blockchain technology can be (or is) applied within HEI is missing (issuing, presenting, verifying credentials)
- The title of Chapter 2 (Materials and Methods) is wrong, and the structure of the chapter is non-existing.
- Moreover, it is not clear what the goal of Chapter 2 is. Supposing that this is as defined in 2.1, i.e., "Analysis of current state-of-the-art of European Digital Education Recognition using blockchain in comparison to best practices in the world" - it is not defined, nor explained how this analysis was conducted (e.g., how and why studies cited were selected) - some references are again cited, without the proper context.
- How do "methods" retaken from [6] fit into the context of the paper? Again, a lot of terms/concepts, without any explanation of how they fit into the context of the paper.
- The "brief blockchain" description given in lines 143-147 was, until this point 3x mentioned in the same manner (problem of paper organization and presentation). Between lines 172-176, already 4th time. Without any relation to anything with this paper.
- What is the goal of the text between lines 167 - 171? What does AI, VR, ML, etc have with the topic of this paper?
- References used in this section are pretty much not recent, so to conclude or identify gaps with them is inappropriate. An objective analysis of the problem needs to be included.
- Outcomes of best practice analysis are, therefore, i.e., because of non-existing (inappropriate/unexisting) state-of-the-art analysis, impossible to be proven, and they may be wrong.
- Chapter 2.2 is fairly written.
The rest of the paper discusses questionary and interviews and their outcomes. As already stated, it is unclear what information if students have the knowledge of/any information about technical terms related to blockchain technology influences the goals of this paper (and how this is related to predefined paper contributions). There is no correlation defined (or proven) in this study between "technical" challenges when adopting BC technology to HEI and the general understanding of terms from BC technology. For example, what is the impact if anyone is aware of tx fees, mining, block reward, hash power, and blockchain fork on applying BC technology to HEI?
- Because of the aforementioned, the paper should not be accepted to be published as a scientific research paper since just analyzing the questionary/interviews without properly applying it to the context of the study (BC technology / HEI) does not make the study scientific.
Author Response

(The authors gave the same response as above.)

Round 2
Reviewer 1 Report
The authors have done a good job of addressing concerns from the first round of reviews and substantially increased the quality of the paper. My only major concern is the language, but that can be addressed with a careful edit (and should be a condition of publication). I would also suggest that the paper would benefit from a better literature review/more citations to existing research as a resources for readers. Aside from that, it is an interesting paper that I will likely cite in the future.
Reviewer 2 Report
The authors have made all the corrections as per the reviewer's comments.